# Complexity of Human Cytomegalovirus Infection in South African HIV-Exposed Infants with Pneumonia

**DOI:** 10.3390/v14050855

**Published:** 2022-04-21

**Authors:** Kerusha Govender, Raveen Parboosing, Salvatore Camiolo, Petr Hubáček, Irene Görzer, Elisabeth Puchhammer-Stöckl, Nicolás M. Suárez

**Affiliations:** 1Department of Virology, University of KwaZulu Natal and National Health Laboratory Service, Durban 4000, South Africa; govenderk7@ukzn.ac.za (K.G.); raveen.parboosing@wits.ac.za (R.P.); 2Medical Research Council-University of Glasgow Centre for Virus Research, Bearsden, Glasgow G61 1QH, UK; salvatore.camiolo@glasgow.ac.uk; 3Department of Medical Microbiology, 2nd Faculty of Medicine, Charles University and Motol University Hospital, 150 06 Prague, Czech Republic; petr.hubacek@fnmotol.cz; 4Center for Virology, Medical University of Vienna, 1090 Vienna, Austria; irene.goerzer@meduniwien.ac.at (I.G.); elisabeth.puchhammer@meduniwien.ac.at (E.P.-S.)

**Keywords:** multiple-strain infections, compartmentalization, human cytomegalovirus, whole genome sequence, HIV, genotype, pneumonia

## Abstract

Human cytomegalovirus (HCMV) can cause significant end-organ diseases such as pneumonia in HIV-exposed infants. Complex viral factors may influence pathogenesis including: a large genome with a sizeable coding capacity, numerous gene regions of hypervariability, multiple-strain infections, and tissue compartmentalization of strains. We used a whole genome sequencing approach to assess the complexity of infection by comparing high-throughput sequencing data obtained from respiratory and blood specimens of HIV-exposed infants with severe HCMV pneumonia with those of lung transplant recipients and patients with hematological disorders. There were significantly more specimens from HIV-exposed infants showing multiple HCMV strain infection. Some genotypes, such as UL73 G4B and UL74 G4, were significantly more prevalent in HIV-exposed infants with severe HCMV pneumonia. Some genotypes were predominant in the respiratory specimens of several patients. However, the predominance was not statistically significant, precluding firm conclusions on anatomical compartmentalization in the lung.

## 1. Introduction

Human cytomegalovirus (HCMV), a ubiquitously distributed betaherpesvirus, usually causes mild or asymptomatic infection in immunocompetent people but can cause significant disease in those with a compromised or an immature immune system (i.e., newborns) [1].

Studies conducted in African populations have shown that a high maternal burden of HIV is associated with a high birth prevalence of HCMV infection [2,3], and this remains a neglected and under-recognized problem [4]. There are also high rates of perinatal and postnatal transmission, largely through breastfeeding [5,6]. The effects of HCMV infection on HIV-exposed infants in Africa include immune dysfunction [7], developmental delay [8,9], rapid HIV disease progression, a higher risk of death [10,11], and end-organ disease manifestations such as gastro-intestinal disease [12] and pneumonia [13,14,15,16]. The precise factors influencing the development of end-organ diseases such as pneumonia are not clear, but it is most likely due to a combination of host and viral factors [17]. In this line, a recent literature review emphasized the significance of HIV exposure and multiple HCMV strains in the development of pneumonia in infants in Africa [18]. 

The large genome size (236 kbp), including at least 170 open reading frames and a natural history that involves superinfection, re-infection, and reactivation present a challenge in elucidating viral genetic markers of clinical significance [19,20]. Certain viral genes, which are thought to be associated with virulence, present a high degree of variability (i.e., hypervariable genes). However, previous studies focusing on the association between viral factors and clinical outcome have examined one or a few of these hypervariable genes [19] and have not found clear associations. 

Another challenge in investigating viral determinants of end-organ disease is the potential compartmentalization of strains [21,22]. A recent study has shown that there is anatomical compartmentalization of viral strains between the cervix and breastmilk, which have significant bearing on HCMV transmission to the infant [23]. The cervical compartment seems to play a role in the transmission bottleneck at birth, and breastmilk is a source of further transmissions resulting in multiple strains in early infancy, especially in countries with high seroprevalence [23,24]. Multiple-strain infections (MSIs) have been associated with poorer outcomes in transplant patients [25], but the clinical impact in other conditions is still unknown. Furthermore, compartmentalization of strains in infants with HCMV end-organ disease such as HCMV pneumonia may have clinical relevance with respect to the diagnosis and treatment of these patients [26]. A previous study assessing the diversity of genes UL74 and UL73 (encoding glycoproteins O [gO] and N [gN], respectively) in blood and lung specimens from HCMV-infected infants in Africa [27] showed similar levels of variability (i.e., genotypes) to those circulating globally. However, the prevalence of other hypervariable genes genotypes may differ geographically, which may have important clinical implications for vaccine design [28].

High-throughput sequencing technologies (HTS) combined with a target enrichment approach has allowed whole HCMV genomes sequencing directly from clinical specimens, enabling a more clinically meaningful investigation [29,30]. This genomic approach has been recently applied to examine HCMV diversity in maternal compartments and suggest that infants in Africa are exposed to multiple strains early in life [23,30]. However, there are currently no genomic studies conducted on samples collected from HCMV-infected infants from Africa with end-organ diseases, which will allow us to further characterize this significant infection.

Here we present an investigation of the HCMV infection complexity in HIV-exposed South African infants with severe HCMV pneumonia by analyzing respiratory and blood specimens using a whole genome approach. We aimed to assess the occurrence of multiple-strain infections and the level of compartmentalization in the lung, while drawing a comparison with HCMV infections in patients with other relevant conditions.

## 2. Materials and Methods

### 2.1. Patients and Specimens

Specimens were obtained between January 2011 and December 2013 in a previous study that included infants from KwaZulu Natal province, South Africa, with confirmed HCMV pneumonia [16]. All patients were HIV-exposed infants with severe pneumonia requiring ventilation and admitted to a pediatric intensive care unit in an academic hospital, Inkosi Albert Luthuli Central Hospital, for management according to the local standard of care. Respiratory and blood specimens were collected early during the hospital admission, prior to commencement of HCMV antiviral therapy. Anonymized blood and non-bronchoscopic bronchoalveolar lavage (NBBAL) specimens were retrieved from −80 °C storage (i.e., a single freeze–thaw cycle). Twenty-four paired specimens from twelve patients were selected (Table 1). Further testing of these specimens was approved by the University of KwaZulu-Natal Biomedical Research Ethics Committee (BE628/16). 

Twenty comparator-paired blood and respiratory specimens were obtained from ten patients who received lung transplants (LT) at the Medical University of Vienna between 2014 and 2017 (ethics approval reference number EK-number 1321/2017). Additional paired blood and respiratory specimens were obtained from seven patients with hematological disorders in Motol University Hospital, Czech Republic, between 2007 and 2016 (ethics approval reference number EK-701c/16).

### 2.2. Nucleic Acid Extraction and HCMV and HIV Testing

The specimens from South Africa were subjected to DNA extraction on the NucliSens easyMAG system (bioMerieux, Marcy-l’Étoile, France) using 200 μL of blood or NBBAL, eluted in 50 µL. Quantitative real-time HCMV PCR was conducted using the LightCycler CMV Quant kit (Roche Diagnostics GmbH, Mannheim, Germany), on the LightCycler 2.0 instrument (Roche^®^, Basel, Switzerland) calibrated to the 1st WHO International Standard for human cytomegalovirus (NIBSC code 09/162) [26]. EDTA blood specimens were tested for HIV using the PCR COBAS^®^ AmpliPrep/COBAS^®^ TaqMan HIV-1 Qualitative test (Roche^®^), as per South African national guidelines at the time of patient recruitment.

The specimens from Austria were assessed for HCMV load during routine diagnosis using the PCR COBAS^®^ AmpliPrep/COBAS^®^ TaqMan CMV Test (Roche^®^). Genomic DNA was extracted from 280 μL of clinical material and eluted in 70 µL of elution buffer using the QIAamp Viral RNA Mini kit (QIAGEN, Hilden, Germany). HCMV quantitation was then performed as described by Kalser et al. [31] using the LightCycler^®^ 480 System (Roche^®^).

Genomic DNA from blood and respiratory specimens from the Czech Republic were extracted using the QIAamp DNA Blood Mini kit and QIAamp DNA Mini kit (QIAGEN), respectively. HCMV quantification was performed on an ABI7500 (Applied Biosystems, Foster City, CA, USA) or a Bio-Rad CFX96 (Bio-Rad Laboratories, Hercules, CA, USA) thermocycler using the method described previously [32].

### 2.3. High-Throughput DNA Sequencing

Sequencing libraries were prepared following the SureSelect version 1.7 target enrichment system (Agilent, Santa Clara, CA, USA) [33]. Briefly, 50 μL of DNA was sheared acoustically using an LE220 sonicator (Covaris, Brighton, UK). Fragmented DNA was end-repaired, A-tailed, and adaptor-ligated using the KAPA LTP library preparation kit (KAPA Biosystems, Wilmington, DE, USA). The resultant HCMV-enriched libraries were then indexed and sequenced on a MiSeq (Illumina, San Diego, CA, USA) with v3 chemistry to yield 300 × 300 nt paired-end reads datasets.

### 2.4. Sequencing Data Analysis, Strain Composition, Genome Assembly, and Compartmentalization

The quality of the datasets was improved by trimming low-quality regions from the reads using Trim Galore v0.4.0 (http://www.bioinformatics.babraham.ac.uk/projects/trim_galore/; length = 21, quality = 10, and stringency = 3, accessed on 9 October 2019). To estimate the sequencing depth of the libraries generated, sequencing reads were aligned against the publicly available sequence of HCMV reference strain Merlin (GenBank accession no. AY446894.2). In addition, dataset fragment diversity was assessed by deduplicating sequencing reads mapped to the strain Merlin sequence, based on alignment coordinates, as described by Suárez et al. [33]. Additionally, the sequencing reads were mapped to all the hypervariable gene genotypes extracted from the complete HCMV genomes deposited in the public domain, and duplicated reads were removed using Picard v2.3.0 (Broad Institute 2019), accessed on 6 May 2021. Deduplicated datasets were genotyped using GRACy [34].

The number of strains represented in a dataset was estimated by a motif read-matching approach described elsewhere [33] (program available at https://centre-for-virus-research.github.io/VATK/HCMV_pipeline, accessed on 9 October 2019). Briefly, trimmed datasets were screened for the presence of short motifs that uniquely identified the different genotypes of 12 HCMV hypervariable genes (RL5A, RL6, RL12, RL13, UL1, UL9, UL11, UL73, UL74, UL120, UL146, and UL139).

In the datasets, in which a single genotype was detected in the 12 hypervariable genes analyzed (i.e., with single HCMV strains), an HCMV genome assembly was attempted using the method described by Suárez et al. [33]. Briefly, for each of these datasets, reads mapping to the Genome Reference Consortium Human Reference 38 sequence (http://genome.ucsc.edu, accessed on 9 October 2019) using Bowtie2 v.2.2.6 [35] were removed. Subsequently, human-depleted datasets were assembled de novo using SPAdes v3.5.0 [36]. The derived contigs were assembled using Scaffold_builder v2.2 [37] and the sequence of HCMV reference strain Merlin to generate HCMV genomes. Finally, human-depleted datasets were mapped to their respective assembled genome using Bowtie2, and the alignments were inspected visually using Tablet v1.21.02.08 [38].

To examine the level of compartmentalization between the respiratory and the circulatory systems, the genotype frequencies of 16 HCMV genes (adding UL20, UL55, UL75, and UL100 to the list of hypervariable genes listed above) were compared between compartments within each patient.

### 2.5. Statistical Analysis

SPSS v26 was used to apply independent *t*-tests and Fisher’s exact or chi-square tests, with a p value of less than 0.05 considered significant.

## 3. Results

### 3.1. Specimen Data

A total of 57 blood and respiratory specimens from 29 patients were analyzed. There were paired respiratory and blood specimens for all except patient VIE04 (whose blood specimen failed during library preparation, possibly due to a low starting viral load). There were twelve HIV-exposed South African infants with HCMV pneumonia, of whom seven were HIV-infected. There were seven patients with hematological disorders from Czech Republic and ten patients from Austria, with various indications for LT (Table 1). The HCMV viral load in the specimens ranged from 1.2 × 103 to 5.9 × 106 genome copies/mL. An independent *t*-test showed that the mean log-transformed HCMV viral load in South African patients did not differ significantly from that in patients from Austria and Czech Republic combined (*p* = 0.428), nor between respiratory and blood specimens (*p* = 0.087).

### 3.2. Quality Assessment of Sequencing Libraries, and Strain Enumeration

A total of 61 sequencing libraries from 57 specimens were generated (i.e., four specimens were repeated), with input HCMV genome copies ranging from 1.92 × 102 to 5.92 × 105 per library (Appendix A, row 10). Sequencing yielded 3.06 × 106 to 2.48 × 107 trimmed reads per library, with 0.2% to 89.1% mapping to the HCMV reference strain Merlin genome (Appendix A, rows 13 and 15). The average sequencing depth, when mapping the trimmed reads to the Merlin reference genome, ranged from 16 to 12,596 reads per nucleotide and covered 68% to 99% of the genome (Appendix A, rows 16 and 17). When mapping deduplicated reads to the Merlin reference genome, the sequencing depth ranged from 1 to 2100 (Appendix A, row 18). 

Of the 12 HIV-exposed South African infants with HCMV pneumonia, 9 were infected with two or more strains in both blood and respiratory specimens, while 3 had evidence of a single-strain infection (Appendix A, row 19).

Of the 17 hematological and LT patients from Czech Republic and Austria, 7 had evidence of multiple-strain infections (MSIs) with up to five strains in either or both compartments, and 10 had single strain infections. 

The propensity for MSIs in the cohort of HIV-exposed infants (mode: two strains) was significantly higher than the cohorts of hematological patients and LT recipients (mode: one strain). A total of 18 of 24 specimens (75.0%) from South Africa had MSI compared with 15 of 36 (41.7%) of non-South African specimens (Fisher’s exact test *p* = 0.017). There was no detectable difference in multiplicity of strains between the respiratory and blood specimens (Fisher’s exact test *p* = 0.548), and the number of strains was not associated with the specimen HCMV viral load (independent *t*-test *p* = 0.731).

### 3.3. HCMV Genomes Assembled

Nine complete HCMV genome sequences were assembled. Complete genome sizes ranged from 234,868 to 236,591 bp. Three genomes presented mutations leading to premature termination of one gene (UL9 in one genome and RL5A in two genomes) (Appendix A, rows 20 and 22).

### 3.4. Analysis of Genotype and Variant Differences between the Cohorts

Those genotypes detected in each specimen with more than 10 reads and at a prevalence of greater than 2% of all reads for that gene locus were considered (Appendix A, rows 25 to 146). We compared the occurrence of each genotype in the cohort of HIV-exposed infants with the cohorts of hematological patients and LT recipients. Of note are the differences in the genotype frequencies in six glycoprotein-encoding genes between the cohorts (Table 2). The expected linkage between UL73 (gN) and UL74 (gO) genes was represented by a high prevalence of the genotypes UL73 G4B and UL74 G4 in the cohort of HIV-exposed infants, whereas they were absent in the other cohorts. We also found significant differences (Fisher’s exact test *p* = 0.001) between the distribution of US9 gene variants. The cohort of HIV-exposed infants presented a higher prevalence of US9 variants with deletions, whereas US9 wild type was more prevalent in the cohorts of hematological patients and LT recipients (Appendix A, rows 149 to 151).

### 3.5. Compartmentalization of Genotypes

HTS is highly PCR-dependent, which might lead to misestimation of genotype frequencies. Consequently, only high-quality datasets—that is, those with (1) an average sequencing depth of unique reads of ≥10 reads per nucleotide and (2) ≥95% coverage of the Merlin strain genome—were considered for compartmentalization analyses. In addition, reads corresponding to hypervariable genes were deduplicated as described in Materials and Methods.

A total of 20 (83%) out of 24 datasets from HIV-exposed infants and 18 (49%) out of 37 datasets from patients with hematological disorders and LT recipients met the defined high-quality criteria (Appendix A). Genotype frequencies were graphically represented in a heatmap (Figure 1). 

Considering the 16 genes interrogated, there was a total of 127 possible genotypes. Of all 127 genotypes, 104 (81.9%) were present in the cohort of HIV-exposed infants, 58 (45.7%) were present in the cohort of lung transplant recipients, and 86 (67.7%) were present in the cohort of patients with hematological disorders.

A total of 4 of the 127 possible genotypes were present in the respiratory but not in the blood specimens in at least three patients—namely, RL13-G1 (patients PRA02, PRA03, and DUR05), UL1-G1 (patients DUR03, DUR04, and DUR05), UL9-G4 (patients PRA03, DUR03, and DUR05), and UL75-G1 (patients PRA04, DUR03, and DUR11).

When analyzing the frequencies of each genotype in each compartment, a predominance of certain genotypes in respiratory specimens was observed (Table 3). However, the differences in genotype frequencies were not statistically significant.

## 4. Discussion

Whole genome sequencing of HCMV enables the possibility of investigating comprehensively the genetic diversity and complexity and the clinical implications of this viral infection [39]. To our knowledge, this is the first study applying a whole genome approach in HIV-exposed infants from South Africa with HCMV pneumonia. Our data show that regardless of HIV infection status and the compartment investigated, HIV-exposed infants with HCMV pneumonia had a high proportion of MSIs (75.0%). These findings are in line with those from a previous study in HIV-exposed infants in sub-Saharan Africa [27]. However, in contrast with our observations, they found MSIs only in samples from the lung. This discrepancy may be a result of the application of different technical approaches with different sensitivities. In addition, the frequency of MSIs observed in our South African cohort is comparable with that of HIV-infected mothers from sub-Saharan Africa (64.3%) [30], which may reflect the high HCMV seroprevalence in Africa.

The high incidence of MSIs in HIV-exposed infants is likely to be the result of in utero superinfection, postnatal reinfection of congenitally infected newborns, repeated re-infection through post-partum close contact with mothers (via breastfeeding) or other infected people, or a combination of these scenarios [23,40]—a postulate which requires further longitudinal studies of HIV- and HCMV-exposed newborns. MSIs have been associated with higher HCMV viral loads [41,42] and delayed clearance [42] in transplant patients and HIV positive patients. However, the effect of MSIs on the clinical outcome of HIV-exposed infants is undescribed. Wider use of whole-genome HCMV sequencing in the study of HIV-exposed infants will be important in elucidating the clinical relevance of these infections, and therefore in assisting clinicians in the management of such patients. Nevertheless, the application of this genomic approach and the acquisition of clinically meaningful data require a comprehensive quality assessment, as highlighted in this and previous studies [33].

In our South African cohort, we observed a significantly higher prevalence of particular genotypes of several glycoprotein-encoding genes, such as gN (genotype 4B) and gO (genotype 4). Our findings are in line with a previous study on the breastmilk of sub-Saharan HIV-infected mothers [30]. Contrarily, a study conducted on HIV-exposed infants found similar genotype frequencies on these genes compared with those found globally [27]. In addition, the predominance of RL12-RL13-UL1 (G2-G2-G2) linkage found previously in breastmilk samples [30] was not replicated in our study, which may reflect the role of this set of genes on cellular tropism [43]. A global analysis of glycoprotein polymorphisms and their clinical significance has an obvious paucity of data from Africa [28]. Therefore, further study of larger, geographically linked and unlinked patient cohorts using a whole genome approach is warranted, as geographic differences in circulating genotypes have implications for antiviral and vaccine development. 

We also found a significant difference in the distribution of US9 variants in the different cohorts, with the wild-type US9 predominating in the European cohorts, while deletion variants, affecting the C terminus of US9, predominating in the South African cohort. This gene is thought to have an immune modulatory role, localizing to mitochondria and causing reduction in interferon β production during later phases of viral replication [44], suggesting that there may be host immune factors that account for the difference in US9 variants between the cohorts.

In general, there were no demonstrable viral genetic determinants of lung compartmentalization, probably because of technical (i.e., limited sample size) or biological (i.e., high vascularization of the lungs) reasons. However, we found that within the RL11 membrane glycoprotein family of genes, a greater proportion of the RL13, UL1, and UL9 genotypes G1, G1, and G4, respectively, were predominant in the respiratory specimens in most of the patients studied. RL13 is an important component of epithelial cell tropism and promoter of cell-to-cell spread, and it is especially important under humoral immune pressure [45]. Therefore, this may reflect a differential infection of alveolar epithelial cells between RL13 genotypes. Similarly, we observed a predominance of UL75 genotype G1 in respiratory specimens, especially in the HIV-exposed infants with HCMV pneumonia. UL75 encodes glycoprotein H, which is an essential component of the pentameric complex (gH/gL/pUL128-131) and the gH/gL/gO trimer, which interacts with platelet-derived growth factor receptor alpha (PDGFRα) during entry to human fibroblasts [46,47]. PDGFRα+ fibroblasts are found in high numbers in infant lungs and are needed for alveolarization to occur, a process that continues in the post-natal period [48,49]. Therefore, UL75 genotype G1 may play a role in lung compartmentalization in pediatric populations. However, this aspect deserves further in vivo and in vitro investigation.

Our study presents some limitations. Firstly, the limited sample size was mainly due to the challenges in collecting clinical samples from pediatric cohorts and the lack of suitability of paired samples from our retrospective studies. However, this is the first study conducted on HIV-exposed infants with HCMV end-organ disease, using a genomic approach on paired blood and respiratory specimens allowing for a comprehensive analysis on the complexity of this infection. Secondly, the simultaneous observational study of transplant and hematological cohorts from differing geographic locations was convenient, but it introduced important confounders, such as differing socio-economic status and immunological and age-related differences that affect HCMV transmission, precluding robust interpretations. Nevertheless, the high prevalence of MSI in HIV-exposed infants that was found here represents a significant observation worthy of further investigation of the determinants and outcomes. In addition, specimens from HIV-unexposed infants with no HCMV end-organ disease were not included in our study, which prevents assessment of immunocompetent subjects. However, the invasiveness of lower respiratory tract sampling presents a challenge in collecting this type of specimen for such patients. Thirdly, compartmentalization is a dynamic process and is best described by longitudinal studies [50]. Our specimens offered just a snapshot of the infection complexity in each compartment but provided no additional information on the inflow and outflow of strains between them. Nevertheless, the whole genome approach conducted in our study allowed us to sequence whole HCMV genomes directly from clinical specimens, which is crucial for clinically meaningful analysis and characterization of this important infection [33]. Furthermore, our study has provided the first complete HCMV genomes circulating in South Africa. Lastly, our study did not include confirmed congenital infections, and therefore we cannot differentiate the scenarios leading to multiple-strain infections. Specimens taken at birth and in early infancy would allow a better description of the relationship between strains in the lung and blood compartment over time and would describe both congenital and perinatal transmission. Additionally, the investigation of HIV-unexposed infants would be essential to clearly elucidate the role of immunocompromise in the acquisition of strains. Moreover, antiviral use would add a further dimension to this type of study and could be conducted with the methods used here, as demonstrated in previous studies [51].

In the future, infants in Africa will clearly benefit from preventing mother-to-child transmission of HIV. However, the continued high burden of HIV in women of child-bearing age [52] highlights the importance of monitoring HCMV infection in HIV-exposed infants.

## 5. Conclusions

We have characterized the complexity of HCMV infection, focusing on the occurrence of multiple-strain infections and compartmentalization in HIV-exposed infants with severe pneumonia. To this end, we applied a whole genome approach on paired blood and respiratory specimens and placed our findings in context with other clinical conditions. We confirmed previous findings of the high incidence of multiple-strain infections in infants in Africa and added to current knowledge on the geographic variation in HCMV strains. We postulate that this finding is clinically significant and warrants longitudinal study measuring clinical outcomes in infants who acquired multiple strains early in life. We have identified several HCMV glycoprotein-encoding gene genotypes that are potential viral genetic determinants of lung compartmentalization. However, this would require further in vitro investigation and longitudinal studies in larger, more geographically diverse cohorts including immunocompetent individuals.

## Figures and Tables

**Figure 1 viruses-14-00855-f001:**
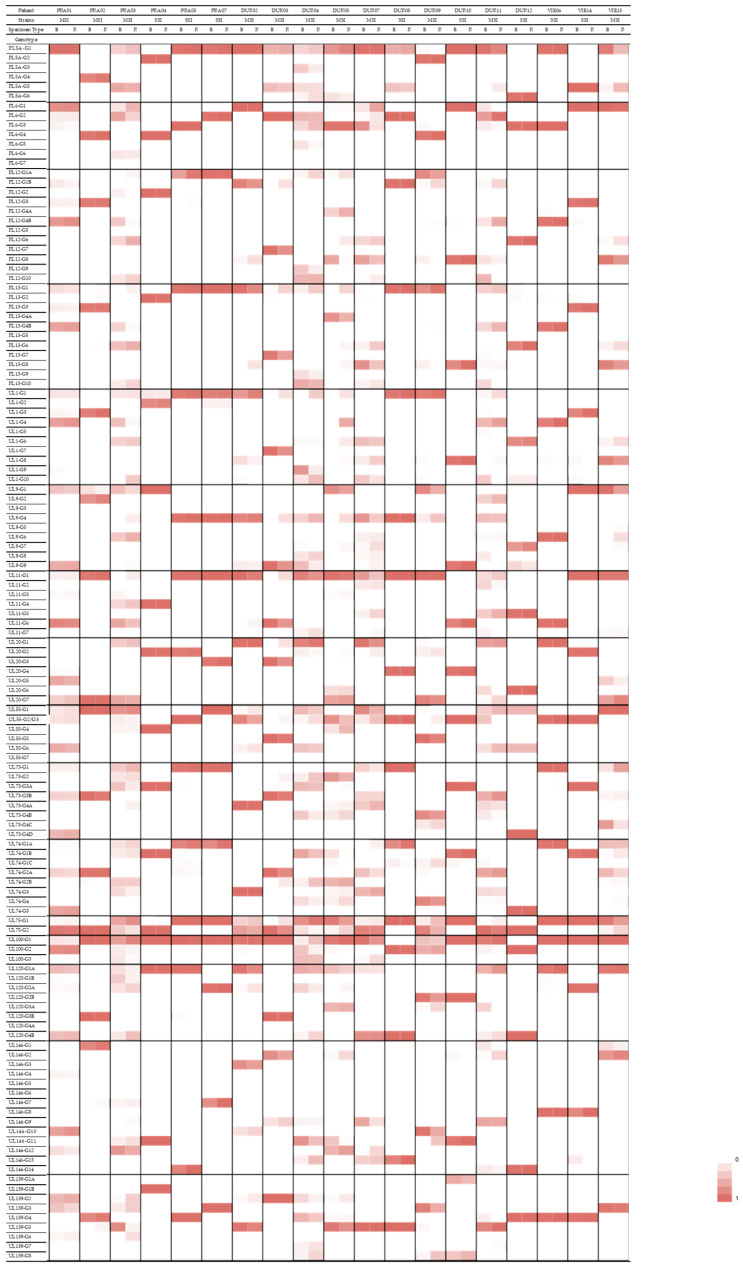
Heatmap showing distribution of genotypes between respiratory and blood specimens. The frequencies of reads are depicted for genotypes in respiratory and blood specimens of eleven patients with multiple-strain infection on a color scale, with darker shading in increments of 20%. MSI = multiple-strain infection. SSI = single-strain infection, B = blood; R = respiratory.

**Table 1 viruses-14-00855-t001:** Patient data.

Patient Code	Patient Location	Patient Diagnosis	Patient Age	HIV Status	Days Post-Transplant	Donor/Recipient HCMV Serostatus	Blood HCMV Load (Genome Copies/mL)	Respiratory HCMV Load (Genome Copies/mL)	Respiratory Specimen Type
DUR01	South Africa	HIV-exposed infants with HCMV pneumonia ^a^	4 months	Negative	N/A	N/A	48,600	11,615	NBBAL
DUR02	South Africa		2 months	Negative	N/A	N/A	26,487	10,230	NBBAL
DUR03	South Africa		4 months	Negative	N/A	N/A	80,676	981,721	NBBAL
DUR04	South Africa		4 months	Positive	N/A	N/A	71,199	2,019,329	NBBAL
DUR05	South Africa		2 months	Negative	N/A	N/A	34,263	181,278	NBBAL
DUR06	South Africa		7 months	Positive	N/A	N/A	7363	58,320	NBBAL
DUR07	South Africa		3 months	Positive	N/A	N/A	58,806	84,564	NBBAL
DUR08	South Africa		3 months	Positive	N/A	N/A	8942	63,909	NBBAL
DUR09	South Africa		2 months	Negative	N/A	N/A	49,572	24,276	NBBAL
DUR10	South Africa		7 months	Positive	N/A	N/A	32,562	132,435	NBBAL
DUR11	South Africa		3 months	Positive	N/A	N/A	35,721	2,964,599	NBBAL
DUR12	South Africa		5 months	Positive	N/A	N/A	826,200	777,600	NBBAL
PRA01	Czech Republic	AML	52 years	Negative	N/A	N/A	5,900,000	867,500	BAL
PRA02	Czech Republic	HSCT	62 years	Negative	67	D−/R+	246,500	55,250	BAL
PRA03	Czech Republic	HSCT	58 years	Negative	62	D+/R+	133,000	672,500	ETA
PRA04	Czech Republic	HSCT	21 years	Negative	227	D−/R+	5,537,500	893,750	BAL
PRA05	Czech Republic	MDS	21 years	Negative	N/A	N/A	903,750	785,000	ETA
PRA06	Czech Republic	Hyper IgM syndrome	8 months	Negative	N/A	N/A	20,750	198,500	ETA
PRA07	Czech Republic	HSCT	58 years	Negative	252	D−/R+	56,750	980,000	BAL
VIE04	Austria	COPD	49 years	Negative	183	D+R+	6630	55,400	BAL
VIE06	Austria	Pulmonary hypertension	32 years	Negative	169	D+/R−	11,300	1620	BAL
VIE08	Austria	COPD	46 years	Negative	187	D−/R+	3390	3650	BAL
VIE13	Austria	Pulmonary fibrosis	50 years	Negative	206	D−/R+	24,800	101,000	BAL
VIE14	Austria	COPD	52 years	Negative	219	D+R+	16,400	132,000	BAL
VIE15	Austria	Pulmonary fibrosis	43 years	Negative	167	D−/R+	18,100	62,200	BAL
VIE16	Austria	COPD	56 years	Negative	456	D+R+	12,300	2380	BAL
VIE17	Austria	Alpha-1 antitrypsin deficiency	54 years	Negative	55	D+R−	1310	28,900	BAL
VIE21	Austria	COPD	57 years	Negative	185	D+R−	1200	11,900	BAL
VIE24	Austria	Bronchiectasis	42 years	Negative	176	D+R+	35,800	134,000	BAL

^a^ Those specimens with a volume greater than 500 μL and HCMV quantification at greater than 3.6 log IU/mL and 4 log IU/mL in blood and NBBAL, respectively, were selected for the present study (cut-off values defined previously (Govender et al. 2017)). AML, acute myeloid leukemia; HSCT, hematopoietic stem cell transplant; MDS, myelodysplastic syndrome; COPD, chronic obstructive pulmonary disease; N/A, not applicable; NBBAL, non-bronchoscopic bronchoalveolar lavage; BAL, bronchoalveolar lavage; ETA, endotracheal aspirate.

**Table 2 viruses-14-00855-t002:** Difference in genotype prevalence between HIV-exposed infants with HCMV pneumonia, compared with patients with hematological disorders (HD) and lung transplant (LT) recipients.

Gene Family/Function	Gene Locus	Genotype	Prevalence (%)		*p*-Value
			HIV-Exposed Infants (South African Cohort) (*n* = 12 Patients)	Patients with HD and LT Recipients (Czech Republic and Austrian Cohorts) (*n* = 17 Patients)	
RL11 family of membrane glycoproteins	UL9	G4	10 (83.3%)	5 (29.4%)	0.008
		G8	4 (33.3%)	0 (0.0%)	0.021
Glycoprotein B	UL55	G2	8 (66.7%)	2 (11.8%)	0.005
		G5	4 (33.3%)	0 (0.0%)	0.021
Glycoprotein N	UL73	G4B	5 (41.7%)	0 (0.0%)	0.007
Glycoprotein O	UL74	G4	5 (41.7%)	0 (0.0%)	0.007
Glycoprotein M	UL100	G2	9 (75.0%)	3 (17.6%)	0.006
Membrane glycoprotein	UL120	G4B	7 (58.3%)	2 (11.8%)	0.014

**Table 3 viruses-14-00855-t003:** Genotypes more predominant in respiratory specimens.

Gene Family/Function	Gene	Genotype	Patients with a Predominance of This Genotype in the Respiratory Specimen
RL11 family of membrane glycoproteins	RL13	G1	PRA02, PRA03, DUR03, DUR04, DUR05, DUR09, DUR11
	UL1	G1	PRA03, DUR02, DUR03, DUR04, DUR05, DUR09, DUR11
	UL9	G4	PRA03, DUR02, DUR03, DUR04, DUR05, DUR09
	UL11	G1	PRA01, PRA02, PRA03, DUR03, DUR09, DUR11
Glycoprotein H	UL75	G1	PRA01, PRA03, DUR02, DUR03, DUR04, DUR07, DUR09, DUR11
Glycoprotein M	UL100	G1	PRA02, PRA03, DUR02, DUR04, DUR05, DUR11, VIE15

## Data Availability

The datasets generated for this study were deposited in the European Nucleotide Archive (ENA; project no. PRJEB45589). The annotated HCMV genome sequences assembled were deposited in ENA (accession numbers: ERZ2490967-75).

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
