# Peer review of "Complexity of Human Cytomegalovirus Infection in South African HIV-Exposed Infants with Pneumonia"

_viruses, 2022, doi:10.3390/v14050855_

Round 1

Reviewer 1 Report

Govender et al have investigated the complexity of HCMV in South Africa using a whole-genome sequencing approach. They found that there were significantly more specimens from HIV-exposed infants showing multiple HCMV strain infections. Furthermore, some genotypes, such as UL73, G4B, and UL74 G4 were significantly more prevalent in HIV-exposed infants with severe HCMV pneumonia.

Overall, this study is technically sound and the manuscript is well written. However, it should be clarified whether multiple-strain infections (MSIs) or detected CMV genotypes in this study are common among immunocompetent individuals or not. Specific comments are as follows.

  1. To investigate the clinical significance of MSIs and CMV genotypes, specimens from immunocompetent individuals with CMV infection should be included. CMV infection is common during infancy, so it would not be so difficult to collect clinical specimens.
  2. Patient age would be an important factor for MSIs because older patients would have more opportunities for re-infection. Therefore, patient age should be included in Table 1.
  3. Figure 1 is so busy.

Author Response

REVIEWER 1

Comments and Suggestions for Authors

Govender et al have investigated the complexity of HCMV in South Africa using a whole-genome sequencing approach. They found that there were significantly more specimens from HIV-exposed infants showing multiple HCMV strain infections. Furthermore, some genotypes, such as UL73, G4B, and UL74 G4 were significantly more prevalent in HIV-exposed infants with severe HCMV pneumonia.

Overall, this study is technically sound and the manuscript is well written. However, it should be clarified whether multiple-strain infections (MSIs) or detected CMV genotypes in this study are common among immunocompetent individuals or not. Specific comments are as follows.

  1. To investigate the clinical significance of MSIs and CMV genotypes, specimens from immunocompetent individuals with CMV infection should be included. CMV infection is common during infancy, so it would not be so difficult to collect clinical specimens.

We agree with Reviewer 1, that specimens from immunocompetent individuals will allow for more robust conclusions, especially on the clinical significance of MSIs. However, obtaining bronchoalveolar lavage from HIV-unexposed infants with no CMV-end-organ disease may have been difficult to justify due to the invasiveness of the specimen collection. Therefore, investigating compartmentalization on the respiratory tract would not have been possible in the immunocompetent individuals. A statement to this effect has been added in lines 328 – 331 and 343 to 345, in the section about limitations, of the revised version of the manuscript.

Although from previous studies, MSIs seem less common in congenital infections, this aspect deserves further investigation, as referred to in lines 339 - 342 of the revised version of the manuscript.

  1. Patient age would be an important factor for MSIs because older patients would have more opportunities for re-infection. Therefore, patient age should be included in Table 1.

Thank you for the suggestion. We have included age in Table 1.

  1. Figure 1 is so busy.

Summarizing genomic data in a concise way is complicated for this type of study. Although we agree with Reviewer 1, we did our best to illustrate as much relevant data as we could in a single figure. The figure, which is available in high resolution, could perhaps be placed over two pages during typesetting, for the details to be more visible to the reader.

Reviewer 2 Report

The manuscript entitled "Complexity of human cytomegalovirus infection in South African HIV-exposed infants with pneumonia". Title, abstract and overall rationale of work to some extent is good. However, there are some major concerns, which needs to be addressed and needs substantial revision.

1) Introduction section: Author need to explain details about the Human cytomegalovirus, HIV and also geographical condition of these kind of diseases. Author must be concise this introduction section in well manner.

2) Material methods section: Author must be elaborate this section specially DNA isolation, how to prepared DNA library, list of primer details and sequencing protocol.  

3) In the results section: There are some query about the results: Author did sequencing of the HCMV and HIV infected infant my query is: Why author is not made venn diagram to show the real picture of this study, How many genes are common between HCMV infected alone, HIV infected alone, both infected HCMV and HIV with comrade with uninfected control group. Author must be make Venn diagram to show clearly results.

4) In African country it is well know that malaria transmission is too high and many studies shown that malaria + HIV infected patient is too high in African country. So my question is that, author confirm that no any other disease found in these infant before collecting the samples.

5) Author must be elaborate the results section in clear way.

6) Figure 1 author need to improve resolution of this figure.

7) I would suggest the authors to enhance your theoretical discussion and arrives your debate or argument.

8) I would like suggest author to summarize the theme of the this article and write the future prospective of this work.

Author Response

REVIEWER 2

Comments and Suggestions for Authors

The manuscript entitled "Complexity of human cytomegalovirus infection in South African HIV-exposed infants with pneumonia". Title, abstract and overall rationale of work to some extent is good. However, there are some major concerns, which needs to be addressed and needs substantial revision.

1) Introduction section: Author need to explain details about the Human cytomegalovirus, HIV and also geographical condition of these kind of diseases. Author must be concise this introduction section in well manner.

This topic has recently been published as a narrative review. We have described this and referenced it in lines 43 – 45 in the revised version of this manuscript.

2) Material methods section: Author must be elaborate this section specially DNA isolation, how to prepared DNA library, list of primer details and sequencing protocol.  

We would like to thank the reviewer for this suggestion.

The detail on DNA isolation is given for each patient cohort, in the section starting on line 109 of the revised manuscript, entitled “ 2.2. Nucleic acid extraction and HCMV and HIV testing”.

Moreover, library preparation and sequencing is described briefly on lines 129 to 132 of the revised manuscript. Further details on library preparation and sequencing are given in reference 33, which is a previous study of our group which focused on the whole genome sequencing method and was published in open-access.

3) In the results section: There are some query about the results: Author did sequencing of the HCMV and HIV infected infant my query is: Why author is not made venn diagram to show the real picture of this study, How many genes are common between HCMV infected alone, HIV infected alone, both infected HCMV and HIV with comrade with uninfected control group. Author must be make Venn diagram to show clearly results.

Thank you for this suggestion. Using a Venn diagram would be an excellent idea for representing absolute presence or absence of a particular genotype/s in the respective cohorts. However, we are also considering the relative differences in prevalence of genotype, thus a Venn diagram will lead us to misinterpret our findings. For example, using a Venn diagram, the genotype UL9 genotype 3 would appear significant as it was present only in the cohort of HIV-exposed infants and in none of the hematological or lung transplant patients. However, this would be a misrepresentation as only 1 HIV-exposed infant had that genotype, giving a non-significant p value. For this reason, we opted to represent the data on a tabulated format (Table 2), as this gave us the possibility to describe the relative differences in prevalence.

4) In African country it is well know that malaria transmission is too high and many studies shown that malaria + HIV infected patient is too high in African country. So my question is that, author confirm that no any other disease found in these infant before collecting the samples.

We thank the reviewer for raising this relevant point. Although malaria is present in our province, there is distinct spatial heterogeneity due to climatic factors. Therefore, clinicians managing infants with sepsis will primarily suspect malaria when there is a history of travel, or residence in the far Northern reaches of our province.  Moreover, this cohort of infants was clinically well-characterized, including a comprehensive description of coinfections, as tabulated and described in reference 16, and none of these patients had malaria infection.

5) Author must be elaborate the results section in clear way.

In relation to this comment, we have edited Table 1 in the results section including the age of participants. Moreover, additional results are included in Supplementary Table 1, as these were too extensive to be included in the main body of the manuscript.

6) Figure 1 author need to improve resolution of this figure.

The lack of resolution of this figure might be associated with the submission process. Although it might look slightly busy, we did our best to illustrate as much relevant data as we could in this figure. We are certain that resolution of this figure will be improved in the published version. In addition, and to facilitate the visualization of this data, this figure could be represented over two pages during typesetting.

7) I would suggest the authors to enhance your theoretical discussion and arrives your debate or argument.

The structure of our discussion section is as follows:

  1. The finding of multiple strain infection in HIV-exposed infants, and how it compares with other conditions and with previous studies
  2. The genotype differences in the South African patients, and how it compares with what others have found
  3. Possible viral genetic determinants of compartmentalization and the scientific plausibility of this.
  4. Limitations of our study and our suggestions to overcome them in future studies.

We have edited the discussion section to further elaborate on the limitations according to the reviewer comments, in lines 328 – 331 and 343 – 345 in the revised manuscript.

We have edited the conclusion in lines 355 – 357 of the revised manuscript to provide a hypothesis generated from the research.

8) I would like suggest author to summarize the theme of the this article and write the future prospective of this work.

To address reviewer suggestion, we have summarized the main findings in lines 352 – 363 of the revised manuscript. Moreover, additional suggestions on future prospects have been included in lines 340 – 346, 348 – 350 and 357 -359 in the revised version of the manuscript.

Round 2

Reviewer 1 Report

The manuscript has been revised well. I think this manuscript will be acceptable.

Reviewer 2 Report

I have completed my evaluation of your manuscript and I found authors have addressed all the concerns raised in the previous version of the manuscript and the quality has improved after incorporating required modifications. Therefore, the manuscript may be considered for publication in this Journal.